# SIRT7 Deficiency Protects against Aβ_42_-Induced Apoptosis through the Regulation of NOX4-Derived Reactive Oxygen Species Production in SH-SY5Y Cells

**DOI:** 10.3390/ijms23169027

**Published:** 2022-08-12

**Authors:** Hironori Mizutani, Yoshifumi Sato, Masaya Yamazaki, Tatsuya Yoshizawa, Yukio Ando, Mitsuharu Ueda, Kazuya Yamagata

**Affiliations:** 1Department of Medical Biochemistry, Faculty of Life Sciences, Kumamoto University, Kumamoto 860-8556, Japan; 2Department of Neurology, Graduate School of Medical Sciences, Kumamoto University, 1-1-1 Honjo, Kumamoto 860-0811, Japan; 3Department of Amyloidosis Research, Faculty of Pharmaceutical Sciences, Nagasaki International University, 2825-7 Huis Ten Bosch Sasebo, Nagasaki 859-3298, Japan; 4Center for Metabolic Regulation of Healthy Aging, Faculty of Life Sciences, Kumamoto University, Kumamoto 860-8556, Japan

**Keywords:** SIRT7, Alzheimer’s disease, amyloid-β, apoptosis, reactive oxygen species, NADPH oxidase

## Abstract

Alzheimer’s disease (AD) is an age-related neurodegenerative disease that is characterized by irreversible memory loss and cognitive decline. The deposition of amyloid-β (Aβ), especially aggregation-prone Aβ_42_, is considered to be an early event preceding neurodegeneration in AD. Sirtuins (SIRT1–7 in mammals) are nicotinamide adenine dinucleotide-dependent lysine deacetylases/deacylases, and several sirtuins play important roles in AD. However, the involvement of SIRT7 in AD pathogenesis is not known. Here, we demonstrate that *SIRT7* mRNA expression is increased in the cortex, entorhinal cortex, and prefrontal cortex of AD patients. We also found that Aβ_42_ treatment rapidly increased NADPH oxidase 4 (NOX4) expression at the post-transcriptional level, and induced reactive oxygen species (ROS) production and apoptosis in neuronal SH-SY5Y cells. In contrast, *SIRT7* knockdown inhibited Aβ_42_-induced ROS production and apoptosis by suppressing the upregulation of NOX4. Collectively, these findings suggest that the inhibition of SIRT7 may play a beneficial role in AD pathogenesis through the regulation of ROS production.

## 1. Introduction

Alzheimer’s disease (AD) is an age-related neurodegenerative disease that is characterized by irreversible memory loss and cognitive decline. AD is the leading cause of dementia and is one of the great healthcare challenges of the 21st century [1]. Senile plaques, which are extracellular deposits of fibrils and amorphous aggregates of amyloid-β (Aβ), are an important pathological feature of AD. Aβ peptides are cleaved products of an amyloid precursor protein (APP) consisting of 36–43 amino acids by the sequential enzymatic action of β-secretase 1 and γ-secretase. Alternatively, APP may be cleaved by α-secretase and γ-secretase to generate non-amyloidogenic peptides. Under normal conditions, Aβ is degraded or eliminated in the cerebral spinal fluid; however, an imbalance between production and clearance causes Aβ to accumulate and aggregate. Although the mechanisms of AD are not fully understood, an excess of Aβ, especially aggregation-prone Aβ_42_, is generally considered to be the initiating step [1,2,3,4,5].

Sirtuins (SIRT1–7 in mammals), evolutionarily conserved nicotinamide adenine dinucleotide (NAD+)-dependent lysine deacetylases/deacylases, are involved in a wide variety of biological processes and age-related diseases [6]. Recent studies also revealed that sirtuins play pivotal roles in neurodegenerative diseases including AD [7,8,9]. For example, SIRT1 reduces the pathological accumulation of Aβ through the activation of the nonamyloidogenic APP processing pathway [10]. SIRT3 plays a neuroprotective role by deacetylating p53; however, SIRT3 expression is decreased in the brain of AD patients [11]. SIRT6 protects neuronal cells from Aβ_42_-induced DNA damage [12]. In contrast, SIRT2 promotes Aβ production by increasing β-secretase 1 expression, and the inhibition of SIRT2 has a beneficial effect on AD mouse models [13]. Thus, sirtuins contribute to AD in multiple ways.

SIRT7 is expressed ubiquitously, including in the brain, and regulates various cellular processes, including the metabolism, inflammation, oncogenesis, and genomic stability. SIRT1 and SIRT6 have beneficial effects against metabolic diseases, but we demonstrate that *Sirt7* knockout mice are protected from high-fat-diet–induced obesity, glucose intolerance, and fatty liver [14,15], suggesting that SIRT7 deficiency plays a beneficial role in metabolic disorders. SIRT1 and SIRT6 exert anti-inflammatory roles by suppressing nuclear factor kappa B (NF-κB). In contrast, SIRT7 promotes inflammation by inhibiting the export of NF-κB from the nucleus [16,17]. In cancer, SIRT1 and SIRT6 act as tumor suppressors [18,19], whereas SIRT7 is responsible for tumor phenotype maintenance, and its expression is increased in many cancers [20,21]. Therefore, SIRT7 and SIRT1/SIRT6 play opposite roles in the metabolism, inflammation, and cancer. However, the involvement of SIRT7 in AD remains unexplored.

In this study, we found that *SIRT7* mRNA expression is upregulated in the cortex of AD patients. Furthermore, we discovered that SIRT7 deficiency prevents Aβ_42_-induced neurotoxicity through the regulation of reactive oxygen species (ROS) production in SH-SY5Y cells. These findings suggest that the inhibition of SIRT7 may play a beneficial role in AD pathogenesis.

## 2. Results

### 2.1. SIRT7 Expression Is Increased in the Brain of AD Patients

Our previous study demonstrated that SIRT7 is expressed widely in the mouse brain [22]. To evaluate the potential roles of SIRT7 in AD, we first assessed the expression profiles of sirtuin genes in the brain of AD patients using previously published datasets (GSE15222, GSE118553, and GSE44770) [23,24,25]. Amyloid deposition occurs in the cortex, entorhinal cortex, and prefrontal cortex during the progression of AD [26,27]. Consistent with a previous report [11], the decreased expression of *SIRT3* mRNA was detected in these regions (Figure 1A). In contrast, *SIRT7* mRNA was increased in these regions of AD patients (Figure 1A–D). These results suggest the involvement of SIRT7 in the pathogenesis of AD.

### 2.2. SIRT7 Knockdown Prevents Aβ_42_-Induced Apoptosis in SH-SY5Y Cells

Aβ oligomerization exerts neurotoxic effects through the activation of caspase 3 [28,29]. To elucidate the role of SIRT7 in AD, we examined Aβ_42_-induced toxicity in *SIRT7* knockdown (KD) neuronal SH-SY5Y cells, a human neuroblastoma cell line (Figure 2A,B). Aβ_42_ oligomer treatment induced cleaved (activated) caspase 3 expression in control SH-SY5Y cells, whereas the activation of caspase 3 was significantly suppressed in *SIRT7* KD SH-SY5Y cells (Figure 2C,D). Immunocytochemical and flow cytometry analyses revealed that Aβ_42_ increased the number of annexin V-positive apoptotic cells in control SH-SY5Y cells (Figure 2E–G). In agreement with the decrease in caspase 3 expression, the number of annexin V-positive apoptotic cells was significantly decreased under the condition of *SIRT7* KD (Figure 2F,G). A lactate dehydrogenase (LDH) assay confirmed that the cytotoxicity of Aβ_42_ oligomers was decreased in *SIRT7* KD SH-SY5Y cells (Figure 2H). Collectively, these results indicate that SIRT7 deficiency inhibits Aβ_42_-induced apoptosis in SH-SY5Y cells.

### 2.3. SIRT7 Controls the Aβ_42_-Induced Increase in Intracellular ROS in SH-SY5Y Cells

Increased ROS production plays a critical role in the onset and progression of AD [2,3,30]. Thus, we examined the effect of Aβ_42_ oligomer treatment on ROS production in SH-SY5Y cells. A marked increase in intracellular ROS, as measured using a DCF-DA probe, was detected following exposure to Aβ_42_ for 3 h, but this increase in ROS was completely repressed by the addition of general antioxidant N-acetyl L-cysteine (NAC) (Figure 3A,B). NAC treatment also inhibited Aβ_42_-induced activated caspase 3 expression (Figure 3C,D) and apoptosis (Figure 3E,F), indicating that the increase in ROS plays a major role in Aβ_42_-induced neurotoxicity in SH-SY5Y cells. We next investigated whether SIRT7 affected Aβ_42_-induced ROS production in SH-SY5Y cells (Figure 3G). Fluorescent staining analysis demonstrated the marked generation of intracellular ROS in control SH-SY5Y cells (Figure 3H). In sharp contrast, ROS production was not detected after Aβ_42_ treatment in *SIRT7* KD cells. Flow cytometry analysis confirmed the inhibition of ROS production by *SIRT7* KD (Figure 3I,J). These results indicate that SIRT7 controls the Aβ_42_-induced elevation of intracellular ROS in SH-SY5Y cells. We then investigated the cellular pathway involved in increased ROS production by Aβ_42_. Mitochondria are an important source of intracellular ROS production [5,31]. However, Aβ_42_ treatment for 3 h did not increase the fluorescence intensity of MitoSOX Red, an indicator of mitochondrial ROS, in either control or *SIRT7* KD SH-SY5Y cells (Figure 3K,L), indicating that mitochondrial ROS are not involved in the increase in Aβ_42_-induced intracellular ROS, at least in these experimental conditions.

### 2.4. NOX4 Contributes to Aβ_42_-Induced ROS Production in SH-SY5Y Cells

Nicotinamide adenine dinucleotide phosphate (NADPH) oxidases (NOXs) are also major sources of ROS. They are multisubunit membrane-bound enzymes that catalyze oxygen reduction into superoxide [32,33]. Intriguingly, Aβ_42_ oligomer-induced ROS production was markedly suppressed by diphenyleneiodonium (DPI), a pan-nhibitor of NOXs, in SH-SY5Y cells (Figure 4A,B). As in the case of NAC treatment, DPI treatment also significantly inhibited Aβ_42_ oligomer-induced caspase 3 activation and apoptosis (Figure 4C–E). These results strongly suggest that NOXs are a major source of Aβ_42_-induced ROS generation and play an important role in cell toxicity. In mammals, the NOX family comprises NOX1–5, dual oxidase 1 (DUOX1), and DUOX2. Among the seven isoforms, NOX4 is expressed in cortical neurons [34]. In line with this, *NOX4* mRNA was almost exclusively expressed in SH-SY5Y cells (Figure 4F). Interestingly, Aβ_42_ treatment for 3 h significantly increased NOX4 protein levels by 2.1-fold without affecting *NOX4* mRNA expression (Figure 4G,H), indicating that Aβ_42_ rapidly regulates NOX4 expression at the post-transcriptional level. Given that NOX4 generates ROS constitutively [35], these findings suggest that NOX4 may contribute to Aβ_42_-induced ROS production and neurotoxicity in SH-SY5Y cells. To this end, we investigated the impact of NOX4 suppression. *NOX4* KD abolished ROS generation by Aβ_42_ in SH-SY5Y cells (Figure 4I–K). *NOX4* KD also significantly suppressed caspase 3 activation (Figure 4L,M) and apoptosis (Figure 4N). These results strongly suggest that NOX4 is a major mediator of Aβ_42_-induced ROS production and apoptosis in SH-SY5Y cells.

### 2.5. SIRT7 Controls Aβ_42_-Induced Apoptosis through the Regulation of NOX4 in SH-SY5Y Cells

To explore the mechanism by which *SIRT7* KD suppresses Aβ_42_-induced neurotoxicity, we investigated the role of SIRT7 in NOX4 expression. Aβ_42_ treatment for 3 h significantly increased NOX4 protein expression without affecting *NOX4* mRNA levels in control SH-SY5Y cells, whereas the increase in protein expression was abrogated by *SIRT7* KD (Figure 5A–C), indicating that SIRT7 deficiency suppresses the upregulation of NOX4 protein levels. We next investigated whether the neuroprotective effects of the loss of SIRT7 are dependent on NOX4 (Figure 5D). Aβ_42_ oligomer-induced ROS production was significantly reduced by *NOX4* KD, as described earlier (Figure 5E,F). Notably, concomitant KD of *SIRT7* and *NOX4* did not result in an additional decrease in ROS production in SH-SY5Y cells compared with *NOX4* KD SH-SY5Y cells (Figure 5F). Similarly, *NOX4* KD resulted in a reduction in caspase 3 activation and the number of apoptotic cells, but there was no additional suppression of caspase 3 activation and apoptosis by concomitant *NOX4* and *SIRT7* KD (Figure 5G–J). Collectively, these results strongly suggest that the inhibition of SIRT7 alleviates Aβ_42_-induced neuronal damage by decreasing the expression of NOX4 (Figure 6).

## 3. Discussion

Previous studies showed that SIRT1, SIRT3, and SIRT6 play protective roles against AD [11,12], but the involvement of SIRT7 in AD remains unknown. Here, we report for the first time that SIRT7 deficiency prevented Aβ_42_-induced neurotoxicity in SH-SY5Y cells. Furthermore, we found that *SIRT7* mRNA expression was upregulated in the cortex, entorhinal cortex, and prefrontal cortex of AD patients. Our findings strongly suggest that SIRT7 and SIRT1/SIRT3/SIRT6 play opposite roles in the pathogenesis of AD, and the inhibition of SIRT7 plays a protective role against AD. The investigation of the roles of SIRT7 knockout in AD mouse models might provide additional information about the contribution of SIRT7 to the pathogenesis of AD.

NOXs are electron-transporting membrane proteins that are responsible for ROS production. Increased NOX activity in the brain of patients with AD and those with mild cognitive impairment was reported [36,37]. A significant correlation between NOX activity and cognitive impairment was also observed in humanized APP × PS1 mice, an animal model of AD [38]. Furthermore, recent studies have revealed that the inhibition of NOX4 by GLX351322, a selective NOX4 inhibitor, protects neuronal cells against Aβ_42_-induced neurotoxicity in APP × PS1 mice [39]. These results strongly support a critical role for NOX4 in AD pathogenesis. Consistent with these findings, we demonstrated that Aβ_42_ oligomer-induced ROS production, caspase 3 activation, and apoptosis were significantly reduced by *NOX4* KD. Moreover, we demonstrated that the short (3 h) exposure of SH-SY5Y cells to the Aβ_42_ oligomer increased NOX4 expression at the post-transcriptional level, whereas SIRT7 deficiency abolished this upregulation. The concomitant KD of *NOX4* and *SIRT7* did not result in a further decrease in ROS production and apoptosis in SH-SY5Y cells compared with *NOX4* KD SH-SY5Y cells. Despite not showing causality, our findings strongly support the notion that SIRT7 deficiency protects against Aβ_42_-induced apoptosis by regulating NOX4 protein levels. Lysine acetylation affects protein stability and protein–protein interactions [40]. Because database analysis (http://pail.biocuckoo.org/index.php, accessed on 30 May 2022) revealed that NOX4 has nine potential lysine acetylation sites, we hypothesized that SIRT7 may control NOX4 expression through its deacetylation. However, this scenario is unlikely, given that no interaction between SIRT7 and NOX4 was detected by a coimmunoprecipitation assay (data not shown). RNA-binding protein ELAV-like protein 1 (ELAVL1) increases NOX4 expression levels by binding to the 3′-untranslated region of *NOX4* mRNA [41]. Therefore, SIRT7 might regulate NOX4 protein expression by deacetylating NOX4-interacting proteins, such as ELAVL1. Further studies are necessary to address how the loss of SIRT7 regulates NOX4 expression levels.

We demonstrated that *SIRT7* mRNA expression was increased in the brain of AD patients. To explore the possibility that Aβ_42_ increases *SIRT7* mRNA expression levels, we examined the effect of Aβ_42_ oligomer exposure on *SIRT7* mRNA expression in SH-SY5Y cells. However, Aβ_42_ oligomer did not affect the levels of *SIRT7* mRNA (data not shown), suggesting that the accumulation of Aβ_42_ is not a direct cause of *SIRT7* upregulation in AD patients. Several studies indicated the occurrence of endoplasmic reticulum (ER) stress in the brain of AD patients [42], and ER stress enhances SIRT7 transcription [43]. Thus, increased ER stress might be involved in the altered expression of SIRT7 in AD patients.

In conclusion, *SIRT7* mRNA expression is upregulated in the cortex of AD patients, and SIRT7 deficiency prevents Aβ_42_-induced neurotoxicity through the regulation of NOX4 expression in SH-SY5Y cells. Further studies are necessary to determine whether inhibition of SIRT7 shows beneficial roles in vivo. In addition, the identification of the exact mechanisms underlying the regulation of NOX4 expression by SIRT7 may lead to a better understanding of the pathogenesis of AD.

## 4. Materials and Methods

### 4.1. Gene Expression Analysis of Human AD Brain

Three expression datasets were used in the analysis. These microarray gene expression datasets from human brain samples (cortex, entorhinal cortex, and prefrontal cortex) were obtained from the Gene Expression Omnibus (GEO) (GEO ID: GSE15222, GSE118553, and GSE44770, respectively). The GSE15222 dataset was composed of expression data from 176 non-AD and 187 AD cases [23]. The GSE118553 dataset consisted of expression data from 27 non-AD and 52 AD cases [24]. The GSE44770 dataset comprised expression data from 101 non-AD and 129 AD cases [25]. All data were analyzed using GEO2R (https://www.ncbi.nlm.nih.gov/geo/geo2r/, accessed on 30 May 2022), and genes of the SIRT family with a *p*-value < 0.05 were considered to be significantly up- or downregulated.

### 4.2. Cell Culture

The SH-SY5Y human neuroblastoma cell line was obtained from the American Type Culture Collection (CRL-2266™). The cells were cultured in a mixed medium of Ham’s F12 and Dulbecco’s Modified Eagle’s Medium (1:1) (11320-033; Gibco™, Thermo Fisher Scientific, Waltham, MA, USA) supplemented with 10% (*v*/*v*) heat-inactivated fetal bovine serum and antibiotics (50 U/mL penicillin and 50 U/mL streptomycin) at 37 °C in a humidified incubator with a 5% CO_2_ atmosphere. The cell medium was replaced every 2 days, and the cells were reseeded after they reached 80% confluence.

### 4.3. Small Interfering RNA (siRNA) Transfection

For *SIRT7* or *NOX4* KD, SH-SY5Y cells were transfected with control scrambled siRNA (4390843; Ambion, Thermo Fisher Scientific) and either human *SIRT7* Silencer Select siRNA (s28303 and s28304; Ambion, Thermo Fisher Scientific) or human *NOX4* Silencer Select siRNA (s224159 and s224160; Ambion, Thermo Fisher Scientific). Transfection was performed with Opti-MEM serum-free medium (31985-070; Gibco™, Thermo Fisher Scientific) and RNAiMAX reagent (13778-150; Thermo Fisher Scientific). Cells (1.0 × 10^6^ cells/well) in a 6-well plate were transfected with a mixture of two types of siRNA (each 5 nM) at a final concentration of 10 nM. The cells were used for experiments after 48 h of siRNA transfection.

### 4.4. Aβ Preparation

Synthetic Aβ_42_ was obtained from the Peptide Institute Inc. (4349-v; Osaka, Japan). Aβ_42_ oligomers were prepared as previously described [44]. In brief, Aβ_42_ peptides were dissolved (2 mM) in anhydrous DMSO, aliquoted, and stored at −80 °C. To yield oligomeric assemblies of Aβ_42_, the stock peptide solution was diluted to 100 μM in phosphate-buffered saline (PBS; 10 mM NaH_2_PO_4_, 137 mM NaCl, 2.7 mM KCl, 1.8 mM KH_2_PO_4_, pH 7.4), immediately vortexed for 30 s, and incubated at 4 °C for 24 h. SH-SY5Y cells were treated with a final concentration of 5 μM Aβ_42_ oligomers according to previous reports [45,46].

### 4.5. Cytotoxicity Assay

Cytotoxicity was determined by the release of LDH in the culture medium. Control and *SIRT7* KD cells (5.0 × 10^4^ cells/well) were seeded in 96-well plates and treated with a 5 μM Aβ_42_-containing medium for 24 h. LDH activity in the culture supernatant was measured using a Cytotoxicity LDH Assay Kit-WST (CK17; Dojindo Molecular Technologies, Rockville, MD, USA). The absorbance of each sample was measured at a wavelength of 490 nm by a microplate reader (Bio-Rad, Hercules, CA, USA).

### 4.6. Immunocytochemical Staining

An Annexin V-Fluorescein Isothiocyanate (FITC) Apoptosis Detection Kit (K101-100; BioVision, Milpitas, CA, USA) and CM-H2DCFHDA (C6827; Invitrogen, Waltham, MA, USA) were used for a cell death assay and cellular ROS imaging, respectively. In brief, SH-SY5Y cells (5.0 × 10^5^ cells/well) were seeded on 35-mm glass-bottom dishes (Greiner Bio-One, Kremsmünster, Austria). After being cultured in medium containing 5 μM Aβ_42_ for 24 h, the cells were washed twice with PBS, and then incubated with annexin V-FITC and propidium iodide (PI) for 5 min at room temperature in the dark. For the detection of ROS, the cells (5.0 × 10^5^ cells/well) were seeded in 35 mm glass-bottom dishes and pretreated with CM-H2DCFHDA (10 μM) for 30 min at 37 °C in the dark, and then treated with 5 μM Aβ_42_ for 3 h. The cells were washed twice with PBS. Fluorescent images were visualized immediately using a confocal laser scanning microscope (BZ-X700; Keyence, Osaka, Japan).

### 4.7. Flow Cytometry Analysis

Flow cytometry was used for the quantitative determination of apoptosis and cellular ROS levels. In brief, SH-SY5Y cells (2.0 × 10^5^ cells/well) were seeded in a 24-well plate and treated the same way as for immunocytochemistry. For mitochondrial ROS detection, the cells were incubated in the presence of 5 μM MitoSOX™ Red (M36008; Invitrogen) for 10 min after Aβ_42_ treatment. After trypsin/EDTA digestion, the medium containing the cells was centrifuged in 1.5 mL tubes (200× *g*, 3 min, 4 °C). The centrifuged cells were resuspended in 2% fetal bovine serum/PBS. Fluorescence intensity was analyzed using a flow cytometer (BD FACS Verse™ Flow Cytometer; BD Biosciences). All experiments were performed in triplicate.

### 4.8. Western Blot Analysis

Western blot analysis was performed as previously described [47]. Total protein lysates of cells were obtained by lysis in RIPA buffer (50 mM Tris-HCl [pH 8.0], 150 mM NaCl, 0.1% sodium dodecyl sulfate, 1% NP-40, 5 mM EDTA, 0.5% sodium deoxycholate, 20 mg/mL Na_3_VO_4_, 10 mM NaF, and 1 mM PMSF) with a protease inhibitor cocktail (04080-11; Nacalai Tesque, Kyoto, Japan). Total protein lysates (60 µg) from SH-SY5Y cells were subjected to gel electrophoresis and protein transfer onto a PVDF membrane (IPVH00010, Immobilon-P; Millipore, Burlington, MA, USA). The following primary antibodies were used: anti-β-actin (clone AC15; Sigma-Aldrich, St. Louis, MI, USA), anti-SIRT7 (clone D3K5A, #5360; Cell Signaling Technology, Danvers, MA, USA), anti-cleaved caspase 3 (#9661; Cell Signaling Technology), and anti-NOX4 (14347-1-AP; Proteintech). After reaction with secondary antibodies, the signals were detected by using Chemi-Lumi One Super Type (Nacalai Tesque) and a ChemiDoc imaging system (BR170-8265; Bio-Rad). All experiments were performed at least three times, and band intensities were quantified using Image Lab Software version 6.0 (Bio-Rad, Hercules, CA, USA).

### 4.9. Treatment of NAC and DPI

To assess the effects of NAC (A9165; Sigma-Aldrich) and DPI (4673-26-1; Cayman Chemical, Ann Arbor, MN, USA), SH-SY5Y cells (1.5 × 10^6^ cells/well) were seeded in a 6-well plate. Cells were preincubated in medium containing 1 mM NAC or 0.1 μM DPI for 1 h, followed by treatment with 5 μM Aβ_42_ in the presence of 1 mM NAC or 0.1 μM DPI for 3 h (ROS assay) or 24 h (apoptosis assay).

### 4.10. Quantitative Real-Time RT-PCR

SH-SY5Y cells were homogenized in Sepasol-RNA I Super G Solution (Nacalai Tesque), and total RNA was isolated using a conventional phenol-chloroform-based RNA extraction method. cDNA was prepared using a PrimeScript RT Reagent Kit and gDNA Eraser (RR047A; TaKaRa Bio, Inc., Shiga, Japan). Quantitative PCR was performed using SYBR Premix Ex Taq II (RR820A; TaKaRa Bio, Inc.) and an ABI 7300 thermal cycler (Applied Biosystems, Foster City, CA, USA; software version 1.4). For each gene, mRNA levels were determined by the comparative cycle threshold (Ct) method (ΔΔCt), and the levels of each mRNA were normalized to ribosomal 18S (*r18S*). The sequences of the primers were as follows. *hr18S*: forward, 5′-GGAGAACTCACTGAGGATGA-3′, reverse, 5′-CCAGTGGTCTTGGTGTGCTG-3′; *hSIRT7*: forward, 5′-ACTTGGTCGTCTACACAGGC-3′, reverse, 5′-CAGACGGGTGATGCTCATGT-3′; *hNOX1*: forward, 5′-CCTGAGTCTTGGAAGTGGATC-3′, reverse, 5′-ACGCTTGTTCATCTGCAATTC-3′; *hCYBB*: forward, 5′-AGCTGAACGAATTGTACGTG-3′, reverse, 5′-ACCCACTATCCATTTCCAAG-3′; *hNOX3*: forward, 5′-TGTGGTCTTGTATGCATGTG-3′, reverse, 5′-CACGCTTTTTCATGTGAAGT-3′; *hNOX4*: forward, 5′-CCAGCTGTACCTCAGTCAAA-3′, reverse, 5′-CCACAACAGAAAACACCAAC-3′; *hNOX5*: forward, 5′-TCTTTCGAGTGGTTTGTGAG-3′, reverse, 5′-ACTTTCTGGAACACCTTGCT-3′; *hDUOX1*: forward, 5′-GTCTTCATGAAAGGCTCTCC-3′, reverse, 5′-AATCTTCCCATGTCAGTTCC-3′; *hDUOX2*: forward, 5′-GACATGGGAGGATTTTCACT-3′, reverse, 5′-CTCGACAGCTGATGTTTTGT-3′.

### 4.11. Statistical Analysis

Results are presented as the mean ± standard error of the mean (SEM) of the indicated number of experiments (*n*). Statistical analysis was performed with an unpaired two-tailed Student’s *t*-test Figure 2B and Figure 4G–I), one-way ANOVA with Tukey’s post hoc test (Figure 5F,H,J), or two-way ANOVA with Tukey’s post hoc test (Figure 2D,G,H, Figure 3B,D,F,J,L, Figure 4B,D,E,K,M,N, and Figure 5B,C). A value of *p* < 0.05 was considered to be statistically significant. All analyses were performed using GraphPad Prism 9 software (GraphPad Software Inc., San Diego, CA, USA; software version 9.4.0).

## 5. Conclusions

SIRT7 deficiency protects against Aβ_42_-induced apoptosis through the regulation of NOX4-derived ROS production in SH-SY5Y cells.

## Figures and Tables

**Figure 1 ijms-23-09027-f001:**
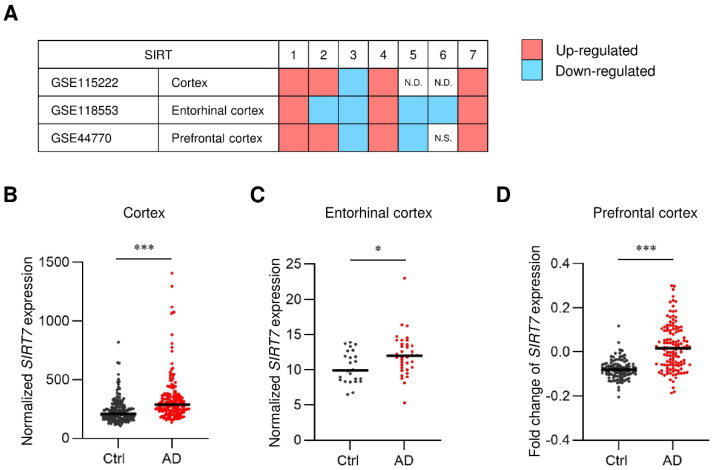
*SIRT7* expression is increased in the brain of AD patients. (**A**) Meta-analysis comparing three public microarray datasets of patients with AD (GSE15222, GSE118553, and GSE44770). The mRNA expression of *SIRT1–7* was compared between AD and control patients. Red and blue boxes represent the upregulation or downregulation of the indicated gene, respectively. N.D., not detected. N.S., not significant. (**B**–**D**) Scatter plot of *SIRT7* mRNA expression in (**B**) cortex from 176 non-AD samples and 187 AD samples (GSE15222), (**C**) entorhinal cortex from 27 non-AD samples and 52 AD samples (GSE118553), and (**D**) prefrontal cortex from 101 non-AD samples and 129 AD samples (GSE44770). Solid lines indicate the median value for each group. All data are shown as the mean ± SEM. Statistical significance was determined by Student’s *t*-test. * *p* < 0.05; *** *p* < 0.001.

**Figure 2 ijms-23-09027-f002:**
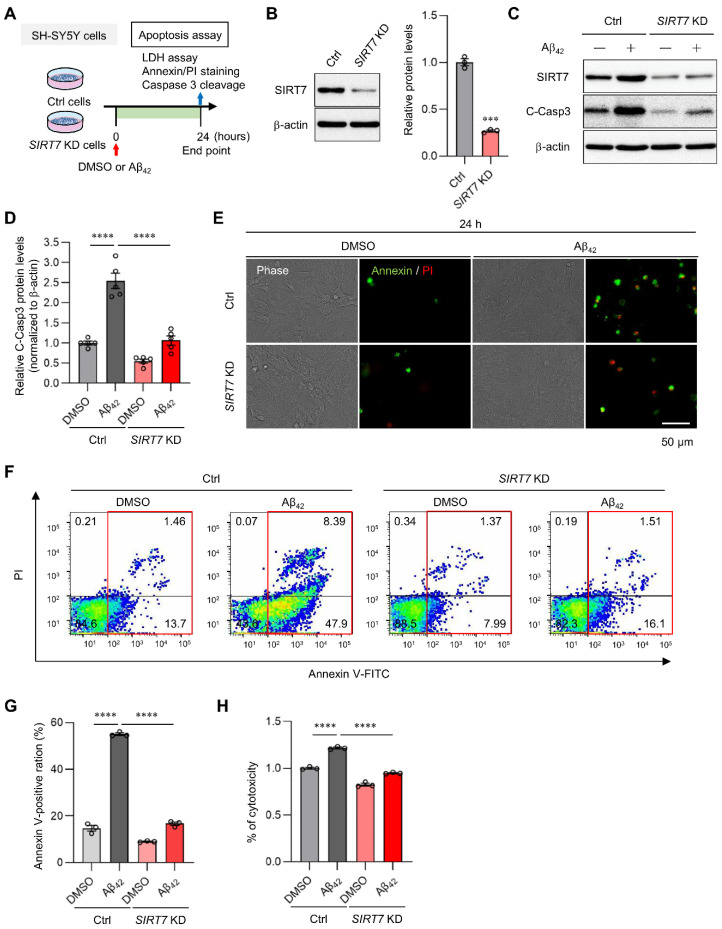
*SIRT7* KD improves Aβ-induced apoptosis. (**A**) Experimental scheme for analyzing Aβ_42_-induced apoptosis in *SIRT7* KD SH-SY5Y cells. (**B**) *SIRT7* KD efficiency was confirmed by Western blot analysis when SH-SY5Y cells were transfected with *SIRT7* siRNA for 48 h. (**C**) After SH-SY5Y cells had been transfected with control and *SIRT7* siRNA for 48 h, cells were treated with 5 μM Aβ_42_ for 24 h. Western blot analysis of cleaved caspase 3 was performed. (**D**) The value of cleaved caspase 3 was normalized to that of β-actin. (**E**) Representative microscopy images of fluorescent annexin V (green)- and PI (red)-stained cells are shown for cells treated in the same condition as that in Figure 2A. Scale bar, 50 μm. (**F**) Flow cytometry analysis was performed on cells treated in the same condition as that in Figure 2A. Representative flow cytometry plots using annexin V-FITC/PI staining for apoptosis. (**G**) Percentage of total annexin V-positive cells was calculated. (**H**) Cell death was evaluated by an LDH activity assay on cells treated in the same condition as in Figure 2A. All data are shown as the mean ± SEM. Statistical significance was determined by either Student’s *t*-test or two-way ANOVA with Tukey’s post hoc test. *** *p* < 0.001; **** *p* < 0.0001.

**Figure 3 ijms-23-09027-f003:**
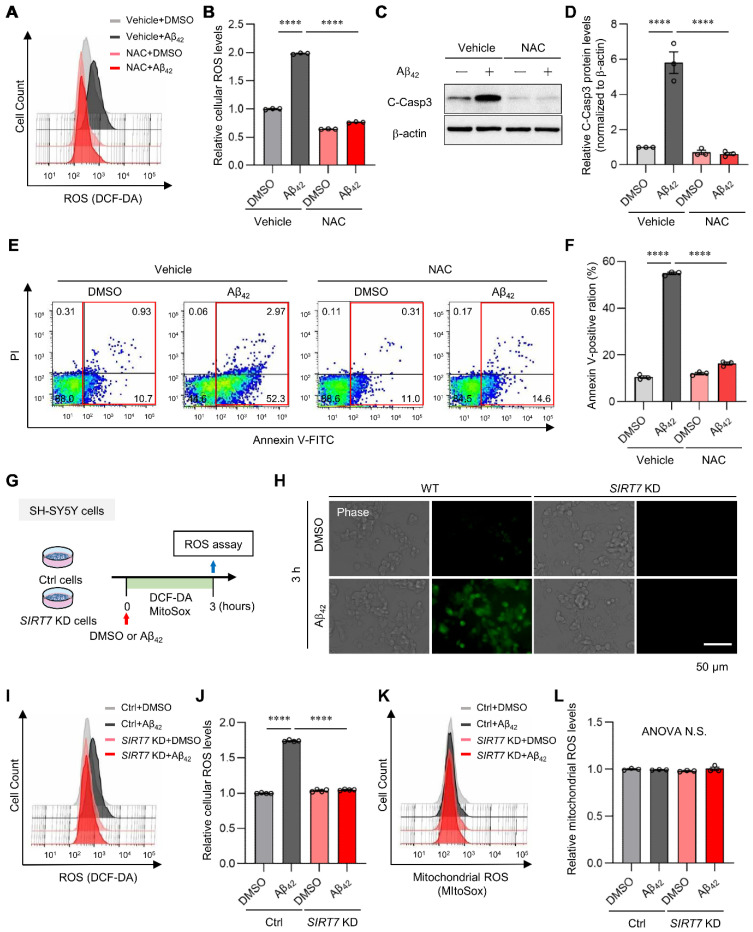
SIRT7 deficiency inhibits Aβ-induced ROS generation. (**A**) Intracellular ROS levels were quantified by flow cytometry after SH-SY5Y cells were loaded with 10 μM CM-H2DCFHDA, pretreated with 1 mM NAC for 1 h, and then treated with 5 μM Aβ_42_ for a further 3 h in the presence of 1 mM NAC. Histogram of DCF-DA intensity of a representative experiment. (**B**) Vertical lines indicate the mean fluorescence values with the control cells set as 1. The geometric mean fluorescence intensity (MFI) ± SEM of three independent experiments was analyzed. (**C**) SH-SY5Y cells were pretreated with 1 mM NAC for 1 h, and then incubated in the presence of 5 μM Aβ_42_ and 1 mM NAC for a further 24 h. The intensity of cleaved caspase 3 was determined by Western blot analysis. (**D**) The value of cleaved caspase 3 was normalized to that of β-actin. (**E**) Flow cytometry analysis was performed on cells treated in the same condition as that in Figure 3C. Representative flow cytometry plots using annexin V-FITC/PI staining for apoptosis. (**F**) The percentage of total annexin V-positive cells was calculated. (**G**) Experimental scheme for analyzing Aβ_42_-induced ROS in *SIRT7* KD SH-SY5Y cells. (**H**) Aβ_42_-induced ROS generation in control and *SIRT7* KD SH-SY5Y cells was evaluated after they had been treated with Aβ_42_ for 3 h. Intracellular ROS levels after DCF-DA loading were visualized by fluorescence microscopy. Scale bar, 50 μm. (**I**) Intracellular ROS levels were quantified by flow cytometry on cells treated in the same condition as that in Figure 3G. Histogram of DCF-DA intensity in a representative experiment. (**J**) The geometric MFI ± SEM of three independent experiments was analyzed. (**K**) Mitochondrial ROS levels were assessed by flow cytometry after the cells had been treated with Aβ_42_ for 3 h and stained with 5 μM MitoSOX Red for a further 15 min. Histogram of MitoSOX Red intensity in a representative experiment. (**L**) The geometric MFI ± SEM of three independent experiments was analyzed. All data are shown as the mean ± SEM. Statistical significance was determined by two-way ANOVA with Tukey’s post hoc test. **** *p* < 0.0001. ANOVA N.S., *p* > 0.05.

**Figure 4 ijms-23-09027-f004:**
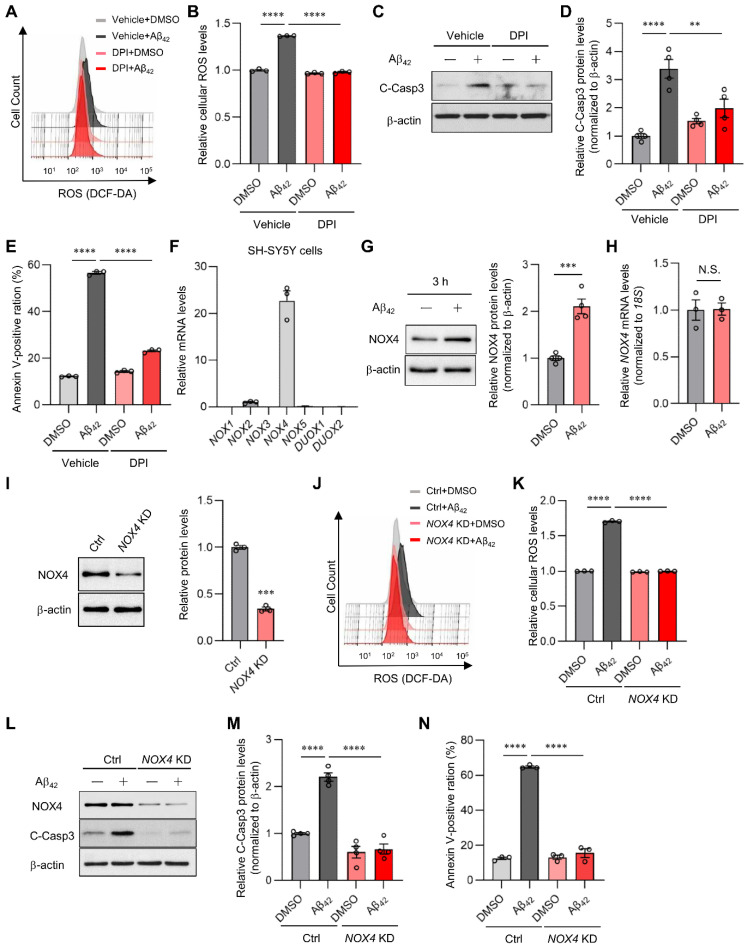
SIRT7 deficiency suppresses NOX-derived ROS generation. (**A**) After 0.1 μM DPI pretreatment for 1 h, SH-SY5Y cells were incubated with 5 μM Aβ_42_ and 0.1 μM DPI for a further 3 h. ROS production was assessed by flow cytometry using DCF-DA. Histogram of DCF-DA intensity in a representative experiment. (**B**) The geometric mean fluorescence intensity (MFI) ± SEM of three independent experiments was analyzed. (**C**) SH-SY5Y cells were pretreated with 0.1 μM DPI and then treated with 5 μM Aβ_42_ for a further 24 h in the presence of 0.1 μM DPI. Cleaved caspase 3 was examined with Western blot analysis. (**D**) The value of cleaved caspase 3 was normalized to that of β-actin. (**E**) Flow cytometry analysis was performed using annexin V-FITC/PI staining to assess apoptosis in cells treated in the same condition as in Figure 4C. The percentages of total annexin V-positive cells were calculated. (**F**) Quantitative RT-PCR analyses were conducted to examine the mRNA levels of the *NOX* family in SH-SY5Y cells. The expression level of the *NOX* family was normalized to that of *18S rRNA*. (**G**) SH-SY5Y cells were incubated with 5 μM Aβ_42_ for 3 h. NOX4 was examined with Western blot analysis. (**H**) *NOX4* mRNA expression was determined by the quantitative real-time RT-PCR analysis of cells treated in the same condition as that in Figure 4G. The value of *NOX4* mRNA was normalized to that of *18S rRNA*. (**I**) *NOX4* KD efficiency was confirmed with Western blot analysis when SH-SY5Y cells were transfected with *NOX4* siRNA for 48 h. (**J**) Intracellular ROS levels were evaluated by flow cytometry after control, and *NOX4* KD SH-SY5Y cells were treated with Aβ_42_ for 3 h. Histogram of DCF-DA intensity of a representative experiment. (**K**) For the quantification of intracellular ROS levels, the geometric MFI ± SEM of three independent experiments was analyzed. (**L**) After SH-SY5Y cells had been transfected with control and *NOX4* siRNA for 48 h, the cells were treated with Aβ_42_ for 24 h. Cleaved caspase 3 protein was evaluated with Western blot analysis. (**M**) The value of cleaved caspase 3 was normalized to that of β-actin. (**N**) Flow cytometry analysis was performed using annexin V-FITC/PI staining to assess apoptosis in cells treated in the same condition as that in Figure 4L. The percentage of total annexin V-positive cells was calculated. All data are shown as the mean ± SEM. Statistical significance was determined by either Student’s *t*-test or two-way ANOVA with Tukey’s post hoc test. N.S., not significant; ** *p* < 0.01; *** *p* < 0.001; **** *p* < 0.0001.

**Figure 5 ijms-23-09027-f005:**
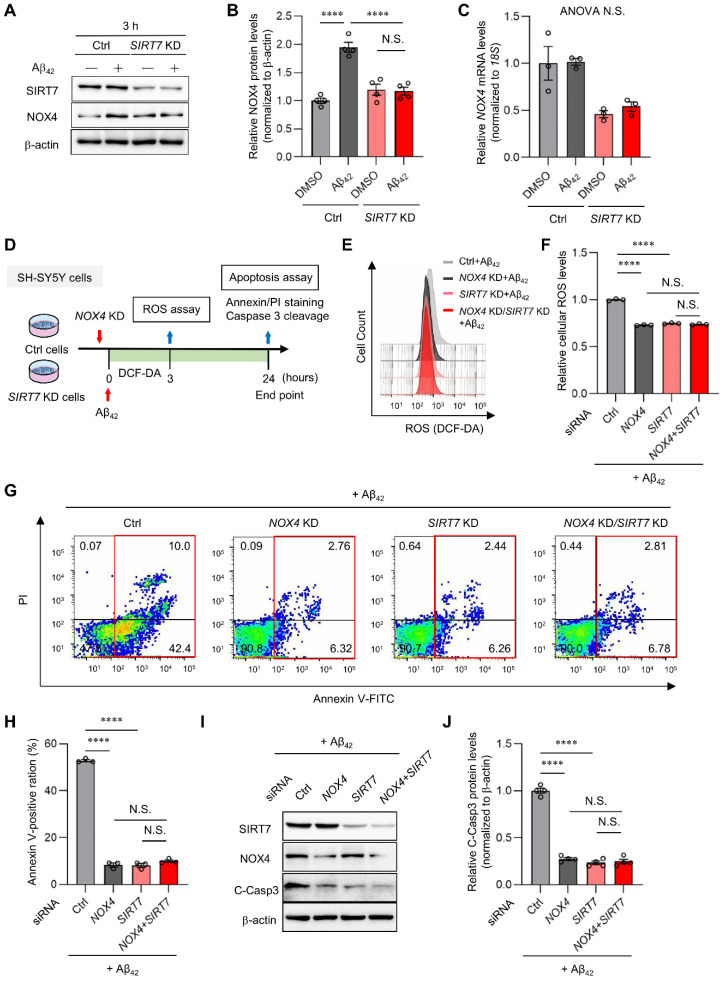
SIRT7 deficiency inhibits Aβ-induced NOX4 protein expression. (**A**) Western blot analysis of NOX4 was performed when control and *SIRT7* KD SH-SY5Y cells had been treated with Aβ_42_ for 3 h. (**B**) The value of NOX4 was normalized to that of β-actin. (**C**) *NOX4* mRNA expression was determined with the quantitative real-time RT-PCR analysis of cells treated in the same condition as that in Figure 5A. The value of *NOX4* mRNA was normalized to that of *18S rRNA*. (**D**) Experimental scheme for the effect of double KD of *SIRT7* and *NOX4* on Aβ_42_-induced ROS and apoptosis. (**E**) Control and *NOX4* KD SH-SY5Y cells transfected with either control siRNA or *SIRT7* siRNA were treated in the presence of 5 μM Aβ_42_ for 3 h. Intracellular ROS levels were evaluated by flow cytometry. (**F**) For quantification of intracellular ROS levels, the geometric MFI ± SEM of three independent experiments was analyzed. (**G**) Flow cytometry analysis was performed on cells treated in the same condition as that in Figure 5D. Representative flow cytometry plots using annexin V-FITC/PI staining for apoptosis. (**H**) The percentage of total annexin V-positive cells was calculated. (**I**) Western blot analysis of cleaved caspase 3 was performed on cells treated in the same condition as that in Figure 5D. (**J**) The value of cleaved caspase 3 was normalized to that of β-actin. All data are shown as the mean ± SEM. Statistical significance was determined by either one-way ANOVA with Tukey’s post hoc test or two-way ANOVA with Tukey’s post hoc test. N.S., not significant; **** *p* < 0.0001. ANOVA N.S., *p* > 0.05.

**Figure 6 ijms-23-09027-f006:**
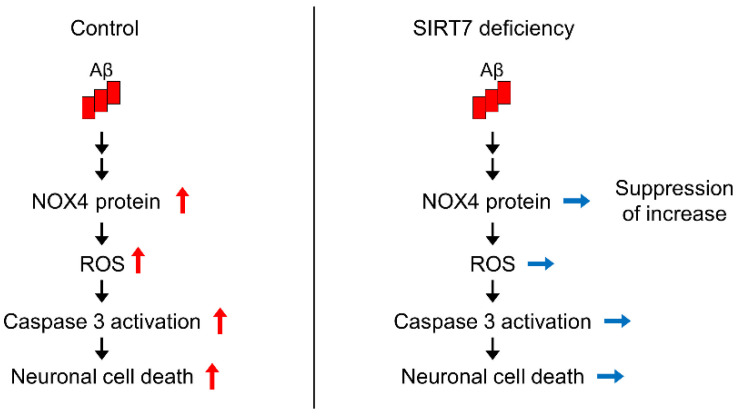
Proposed model of how SIRT7 deficiency protects against Aβ_42_-induced neuronal cell death.

## Data Availability

Not applicable.

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
