# Peer review of "SIRT7 Deficiency Protects against Aβ42-Induced Apoptosis through the Regulation of NOX4-Derived Reactive Oxygen Species Production in SH-SY5Y Cells"

_ijms, 2022, doi:10.3390/ijms23169027_

Round 1

Reviewer 1 Report

Summary

In this study, authors identified upregulated SIRT7 mRNA in cortices of AD patients. Furthermore, using a cell culture model, the authors show that SIRT7 deficiency prevents Aβ42-induced neurotoxicity through the regulation of ROS production in SH-62 SY5Y cells, apparently associated with NOX4, preventing Aβ42-induced apoptosis. The authors propose a protective role for reduced SIRT7 expression as SIRT7 knockdown in SH-SY5Y cells inhibited Aβ42 treatment-mediated apoptosis by suppressing NOX4 protein expression upstream of ROS production.

Strengths:

-        This is a well written, well organized, and well-illustrated article on an interesting subject matter. The methods are appropriate and experimental plan is well described. The data are clear and easy to follow.

Both human AD brain (microarray gene expression datasets obtained from the Gene Expression Omnibus) and SHSY5Y model were used for experiments, which strengthens the manuscript.

Data were analyzed via a combination of methods, incorporating cytotoxicity (LDH) assay, immunohistochemistry staining, flow cytometry technique, and western blotting.

Potential effects of DMSO vehicle are addressed in the manuscript.

Major concern:

1.     The authors have used T test for comparing multiple groups throughput the paper. ANOVA should be used for comparing multiple groups. For instance, in Figure 2D, while comparing levels of cleaved Casp3 – there are 2 factors Aβ42 treatment and SIRT7 knockdown. A significant interaction between the 2 factors is necessary to make the statement that “SIRT7 KD improves Aβ-induced apoptosis”. This is especially important because: i) Aβ42 treatment causes 2.5 fold increase in cleaved Casp3 levels, ii) SIRT7 kd causes a decrease in cleaved Casp3 levels and Aβ42 treatment causes a 2 fold increase in cleaved Casp3 levels. Whether these 2 experimental conditions are significantly different can be assessed only by ANOVA. 

Minor concern.

1.     In Figure 2D, Aβ42 treatment is associated with 2.5-fold increase in cleaved Casp3 level. In Figure 3D, Aβ42 treatment is associated with 6-fold increase in cleaved Casp3 level. In Figure 4D, Aβ42 treatment is associated with 3-fold increase in cleaved Casp3 level. In Figure 4M, Aβ42 treatment is associated with 2-fold increase in cleaved Casp3 level. Thus, there is variability in the cleaved Casp3 level. This may possibly be due to the loading controls being run separately. Request that the authors please clarify this.

2. The authors state that for flow cytometry, “Each experiment was repeated at least three times.” Were triplicates performed for each experimental set? If not, please address.

3. For Immunohistochemistry methods, authors state that “Fluorescent images were visualized immediately using a confocal laser scan-338 ning microscope (BZ-X700; Keyence)”, but this does not clarity how quantitation of data was accomplished, e.g., was there nonbiased, automated detection of Annexin 488 and PI label by the software used? Or was there manual counting (and in this case how many reviewers were there and what was done to address variability in scoring)? Please specify how the immunofluorescent analysis was performed.

4. Fig 1, key, minor typographical error “Dow n-regulated” should be “Downregulated”

5. 2. The authors suggest that their data may indicate that SIRT7 deficiency suppresses the upregulation of NOX4 protein levels. The data show a potential association. Is there any previously published evidence for this, or suggestion of an interaction? If so, please cite in the manuscript. The discussion section could be slightly elaborated. An association may be causative, or may not. Without a definite mechanism shown, it be perhaps be best to discuss the association and a potential cause-effect.

Thank you.

Author Response

Response to comments of reviewer #1

We thank the reviewer for stating that “This is a well written, well organized, and well-illustrated article on an interesting subject matter” and we appreciate the reviewer’s constructive suggestions. We have substantially revised our manuscript based on these valuable comments.

1) The authors have used T test for comparing multiple groups throughput the paper. ANOVA should be used for comparing multiple groups. For instance, in Figure 2D, while comparing levels of cleaved Casp3 – there are 2 factors Aβ42 treatment and SIRT7 knockdown. A significant interaction between the 2 factors is necessary to make the statement that “SIRT7 KD improves Aβ-induced apoptosis”. This is especially important because: i) Aβ42 treatment causes 2.5 fold increase in cleaved Casp3 levels, ii) SIRT7 kd causes a decrease in cleaved Casp3 levels and Aβ42 treatment causes a 2 fold increase in cleaved Casp3 levels. Whether these 2 experimental conditions are significantly different can be assessed only by ANOVA.

We thank the reviewer for pointing this out and we apologize for our error. In accordance with the reviewer’s comment, we have performed a statistical analysis using ANOVA for comparison of more than two groups (one-way ANOVA: Figures 5F, H, and J; two-way ANOVA: Figures 2D, G, H; 3B, D, F, J, L; 4B, D, E, K, M, N; and 5B, C). The results of the interactions were as follows:

Figure 2D: F (1, 16)=19.76, p=0.0004

Figure 2G: F (1, 8)=479.3, p<0.0001

Figure 2H: F (1, 8)=26.98, p=0.0008

Figure 3B: F (1, 8)=5874, p<0.0001

Figure 3D: F (1, 8)=60.13, p<0.0001

Figure 3F: F (1, 8)=1577, p<0.0001

Figure 3J: F (1, 12)=4529, p<0.0001

Figure 3L: F (1, 8)=2.351, p=0.1637 (not significant)

Figure 4B: F (1, 18)=861.0, p<0.0001

Figure 4D: F (1, 12)=16.12, p=0.0017

Figure 4E: F (1, 18)=2248, p<0.0001

Figure 4K: F (1, 8)=5696, p<0.0001

Figure 4M: F (1, 12)=38.89, p<0.0001

Figure 4N: F (1, 8)=301.7, p<0.0001

Figure 5B: F (1, 12)=41.68, p<0.0001

Figure 5C: F (1, 8)=0.1164, p=0.7418 (not significant)

Because significant interactions were detected between two factors in Figures 2D, G, H; 3B, D, F, J; 4B, D, E, K, M, N; and 5B, we performed multiple comparisons using Tukey’s post hoc test.

We revised the relevant text in the Materials and Methods section as follows: “Statistical analysis was performed with an unpaired two-tailed Student’s t-test (Figures 2B and 4G–I), one-way ANOVA with Tukey’s post hoc test (Figure 5F, H, and J), or two-way ANOVA with Tukey’s post hoc test (Figures 2D, G, H; 3B, D, F, J, L; 4B, D, E, K, M, N; and 5B, C). A value of p < 0.05 was considered to be statistically significant. All analyses were performed using GraphPad Prism 9 software.” (page 15, lines 428 to 432). We also have added the following information in the figure legends: “Statistical significance was determined by either Student’s t-test or two-way ANOVA with Tukey's post hoc test (Figures 2 and 4).”, “Statistical significance was determined by two-way ANOVA with Tukey's post hoc test (Figure 3).” and “Statistical significance was determined by either one-way ANOVA with Tukey's post hoc test or two-way ANOVA with Tukey's post hoc test (Figure 5).”

2) In Figure 2D, Aβ42 treatment is associated with 2.5-fold increase in cleaved Casp3 level. In Figure 3D, Aβ42 treatment is associated with 6-fold increase in cleaved Casp3 level. In Figure 4D, Aβ42 treatment is associated with 3-fold increase in cleaved Casp3 level. In Figure 4M, Aβ42 treatment is associated with 2-fold increase in cleaved Casp3 level. Thus, there is variability in the cleaved Casp3 level. This may possibly be due to the loading controls being run separately. Request that the authors please clarify this.

We apologize for the confusing description in the original manuscript. To clarify, SH-SY5Y cells (1.0 × 106 cells/6-well) were transfected with siRNA for 48 h and then treated with Aβ42 oligomer for additional 24 h before western blotting (Figures 2D and 4M) (page 6, lines 176 to 178, page 10, lines 231 to 233). Meanwhile, another sample of SH-SY5Y cells (1.0 × 106 cells/6-well) was treated with Aβ42 oligomer for 24 h before western blotting (Figures 3D and 4D). The difference in experimental protocols may be a possible cause for the observed difference in relative cleaved caspase 3 protein levels (Figures 2D and 4M: 2.2- to 2.5-fold; Figures 3D and 4D: 3.4- to 5.8-fold). Because the experimental methods for Figures 3D and 4D were not described in the original manuscript, we have added this information to the revised manuscript on page 15, lines 398 to 403 as follows: “Treatment of NAC and DPI: To assess the effects of NAC (A9165; Sigma-Aldrich) and DPI (4673-26-1; Cayman Chemical), SH-SY5Y cells (1.5 × 106 cells/well) were seeded in a 6-well plate. Cells were preincubated in medium containing 1 mM NAC or 0.1 μM DPI for 1 h, followed by treatment with 5 μM Aβ42 in the presence of 1 mM NAC or 0.1 μM DPI for 3 h (ROS assay) or 24 h (apoptosis assay).”

3) The authors state that for flow cytometry, “Each experiment was repeated at least three times.” Were triplicates performed for each experimental set? If not, please address.

We apologize for the poor description in the original manuscript. We performed all flow cytometry experiments in triplicate. We have added this information to the revised manuscript on page 14, lines 380 to 381.

4) For Immunohistochemistry methods, authors state that “Fluorescent images were visualized immediately using a confocal laser scanning microscope (BZ-X700; Keyence)”, but this does not clarity how quantitation of data was accomplished, e.g., was there nonbiased, automated detection of Annexin 488 and PI label by the software used? Or was there manual counting (and in this case how many reviewers were there and what was done to address variability in scoring)? Please specify how the immunofluorescent analysis was performed.

We apologize that for the confusing description of the immunohistochemistry. Quantitative analysis was performed using the data from the flow cytometry experiments (Figures 2G; 3F, I–L; 4A, B, E, J, K, N; and 5E, F, H). As stated in the Figure legends in the original manuscript, we showed only representative microscopy images of annexin/PI positive cells (Figure 2E) and intracellular ROS levels (Figure 3H).

5) Fig 1, key, minor typographical error “Dow n-regulated” should be “Downregulated

We apologize for the typo. We have corrected this accordingly.

6) The authors suggest that their data may indicate that SIRT7 deficiency suppresses the upregulation of NOX4 protein levels. The data show a potential association. Is there any previously published evidence for this, or suggestion of an interaction? If so, please cite in the manuscript. The discussion section could be slightly elaborated. An association may be causative, or may not. Without a definite mechanism shown, it be perhaps be best to discuss the association and a potential cause-effect.

We thank the reviewer for pointing this out and agree with the comment about our original manuscript. To our knowledge, this is the first report that SIRT7 deficiency inhibits NOX4 protein elevation, and there are no reports suggesting a relationship between SIRT7 and NOX4. The results of NOX4/SIRT7 double knockdown experiments strongly suggest that SIRT7 deficiency protects against Aβ42-induced apoptosis through the suppression of NOX4. However, we do not know the exact mechanism underlying the regulation of NOX4 protein expression by SIRT7 deficiency at this point. Therefore, we address this point in the Discussion section of the revised manuscript as follows: “Of note, concomitant KD of NOX4 and SIRT7 did not result in further decrease of ROS production and apoptosis in SH-SY5Y cells compared with NOX4 KD SH-SY5Y cells. Despite not showing causality, our findings strongly support the notion that SIRT7 deficiency protects against Aβ42-induced apoptosis by regulating NOX4 protein levels.” (page 13, lines 281 to 285).

Reviewer 2 Report

Recent studies have revealed that sirtuins play important roles in neurodegenerative diseases including AD. In this study, Mizutani et al. investigated the involvement of SIRT7 in AD remained unknown. The results suggested that SIRT7 deficiency protects against Aβ42-induced apoptosis by regulating NOX4-derived ROS production in SH-SY5Y cells. The experiments are well designed, the results are clear, and the paper is well written. I recommend the acceptance of the publication of this manuscript after the response to the following minor comments.

Minor comments:

(1) There is a space between w and n in "Dow n-regulated" in Figure 1A, so delete it.

(2) Page 10, line 302; please revise letter “2” in CO2 to a subscript.

(3) Please add a description of the quantification of western blot analysis. For example, how many experiments were performed and how many blots were analyzed.

(4) A brief description in Non-published data (original images for blots) will help the reader. For example, explanation of red frames and black frames, meaning of two black frames of C-Casp3 in Figure 3C, two blots below Figure 4C (top is C-Casp3 ?, bottom is β-actin?)

Author Response

Response to comments of reviewer #2

We thank the reviewer for stating that “The experiments are well designed, the results are clear, and the paper is well written. I recommend the acceptance of the publication of this manuscript after the response to the following minor comment.” We appreciate the reviewer’s constructive suggestions. We have substantially revised our manuscript based on these valuable comments.

There is a space between w and n in "Dow n-regulated" in Figure 1A, so delete it.

We apologize for the typo. We have corrected this accordingly.

2) Page 10, line 302; please revise letter “2” in CO2 to a subscript.

We apologize for the formatting oversight. We have changed this accordingly.  

3) Please add a description of the quantification of western blot analysis. For example, how many experiments were performed and how many blots were analyzed.

We thank the reviewer for pointing this out. In the revised manuscript, we have added the following text to the Materials and Methods section: “All experiments were performed at least three times and band intensities were quantified using Image Lab software (Bio-Rad).” (page 15, lines 395 to 396).

4) A brief description in Non-published data (original images for blots) will help the reader. For example, explanation of red frames and black frames, meaning of two black frames of C-Casp3 in Figure 3C, two blots below Figure 4C (top is C-Casp3 ?, bottom is β-actin?)

We apologize that our inadequate explanation in the original manuscript. We have added the protein name to the figure and the following explanation to the section on non-published data: “Red boxes indicate representative data, and black boxes indicate data used to quantify band intensities.”

Reviewer 3 Report

In this work, the authors performed a thorough investigation on whether the regulation of SIRT7 gene could have an impact on the Aβ42-induced apoptosis using a SH-SY5Y cell model. They found that the inhibition of SIRT7 may have a protective role in AD pathogenesis through the regulation of ROS production through the regulation of NOX4-derived reactive oxygen. They provide detailed analysis using a variety of biological methods and well-planned control experiments. The manuscript is well-written and I did not find any major problems. In general, this manuscript can be published in IJMS. Yet, I do suggest the authors consider the following comment in their revised manuscript.

The manuscript should provide more discussions in the introduction section to highlight the importance of SIRT7. The introduction currently appears to focus on the discussions about the roles of SIRT1, SIRT2, SIRT3, SIRT6.

Author Response

We thank the reviewer for stating that “The manuscript is well-written and I did not find any major problems. In general, this manuscript can be published in IJMS.” We appreciate the reviewer’s constructive suggestions. We have substantially revised our manuscript based on these valuable comments.

The manuscript should provide more discussions in the introduction section to highlight the importance of SIRT7. The introduction currently appears to focus on the discussions about the roles of SIRT1, SIRT2, SIRT3, SIRT6.

We thank the reviewer for pointing this out and agree with the comment about our original manuscript. In the revised manuscript, we added the following explanation of SIRT7 to the Introduction: “SIRT1 and SIRT6 have beneficial effects against metabolic diseases, but we demonstrated that Sirt7 knockout mice are protected from high-fat-diet–induced obesity, glucose intolerance, and fatty liver [Yoshizawa T. 2014, Yamagata K. 2018], suggesting that SIRT7 deficiency has beneficial roles in metabolic disorders. SIRT1 and SIRT6 exert anti-inflammatory roles by suppressing nuclear factor kappa B (NF-kB). In contrast, SIRT7 promotes inflammation by inhibiting the export of NF-kB from the nucleus [Miyasato Y. 2018, Sobuz SU. 2019]. In cancer, SIRT1 and SIRT6 act as tumor suppressors [Chang HC. 2014, Tasselli L. 2017], whereas SIRT7 is responsible for tumor phenotype maintenance and its expression is increased in many cancers [Barber MF. 2012, Wu D. 2018].” (page 2, lines 59 to 66).

References

Barber, M.F.; Michishita-Kioi, E.; Xi, Y.; Tasselli, L.; Kioi, M.; Moqtaderi, Z.; Tennen, R.I.; Paredes, S.; Young, N.L.; Chen, K.; et al. SIRT7 Links H3K18 Deacetylation to Maintenance of Oncogenic Transformation. Nature 2012, 487, 114–118, doi:10.1038/nature11043.

Chang, H.C.; Guarente, L. SIRT1 and Other Sirtuins in Metabolism. Trends Endocrinol. Metab. 2014, 25, 138–145, doi:10.1016/j.tem.2013.12.001.

Miyasato, Y.; Yoshizawa, T.; Sato, Y.; Nakagawa, T.; Miyasato, Y.; Kakizoe, Y.; Kuwabara, T.; Adachi, M.; Ianni, A.; Braun, T.; et al. Sirtuin 7 Deficiency Ameliorates Cisplatin-Induced Acute Kidney Injury Through Regulation of the Inflammatory Response. Sci. Rep. 2018, 8, 1–14, doi:10.1038/s41598-018-24257-7.

Sobuz, S.U.; Sato, Y.; Yoshizawa, T.; Karim, F.; Ono, K.; Sawa, T.; Miyamoto, Y.; Oka, M.; Yamagata, K. SIRT7 Regulates the Nuclear Export of NF-ΚB P65 by Deacetylating Ran. Biochim. Biophys. Acta - Mol. Cell Res. 2019, 1866, 1355–1367, doi:10.1016/j.bbamcr.2019.05.001.

Tasselli, L.; Zheng, W.; Chua, K.F. SIRT6: Novel Mechanisms and Links to Aging and Disease. Trends Endocrinol. Metab. 2017, 28, 168–185, doi:10.1016/j.tem.2016.10.002.

Wu, D.; Li, Y.; Zhu, K.S.; Wang, H.; Zhu, W.G. Advances in Cellular Characterization of the Sirtuin Isoform, SIRT7. Front. En-docrinol. (Lausanne). 2018, 9, 652, doi:10.3389/fendo.2018.00652.

Yamagata, K.; Yoshizawa, T. Transcriptional Regulation of Metabolism by SIRT1 and SIRT7. Int. Rev. Cell Mol. Biol. 2018, 335, 143–166, doi:10.1016/bs.ircmb.2017.07.009.

Yoshizawa, T.; Karim, M.F.; Sato, Y.; Senokuchi, T.; Miyata, K.; Fukuda, T.; Go, C.; Tasaki, M.; Uchimura, K.; Kadomatsu, T.; et al. SIRT7 Controls Hepatic Lipid Metabolism by Regulating the Ubiquitin-Proteasome Pathway. Cell Metab. 2014, 19, 712–721, doi:10.1016/j.cmet.2014.03.006.